# Faster Online Learning of Optimal Threshold for Consistent F-measure Optimization

**Mingrui Liu**[*†]**, Xiaoxuan Zhang**[*†]**, Xun Zhou**[‡]**, Tianbao Yang**[†]
[†]Department of Computer Science, The University of Iowa, Iowa City, IA 52242, USA
[‡]Department of Management Sciences, The University of Iowa, Iowa City, IA 52242, USA
`mingrui-liu, tianbao-yang@uiowa.edu`

## Abstract

In this paper, we consider online F-measure optimization (OFO). Unlike traditional performance metrics (e.g., classification error rate), F-measure is non-decomposable over training examples and is a non-convex function of model parameters, making it much more difficult to be optimized in an online fashion. Most existing results of OFO usually suffer from high memory/computational costs and/or lack statistical consistency guarantee for optimizing F-measure at the population level. To advance OFO, we propose an efficient online algorithm based on simultaneously learning a posterior probability of class and learning an optimal threshold by minimizing a stochastic strongly convex function with unknown strong convexity parameter. A key component of the proposed method is a novel stochastic algorithm with low memory and computational costs, which can enjoy a convergence rate of $\widetilde{O}(1/\sqrt{n})$ for learning the optimal threshold under a mild condition on the convergence of the posterior probability, where $n$ is the number of processed examples. It is provably faster than its predecessor based on a heuristic for updating the threshold. The experiments verify the efficiency of the proposed algorithm in comparison with state-of-the-art OFO algorithms.

## 1 Introduction

A learning algorithm is to optimize a certain performance metric defined over a set or population of examples. Online learning [18, 28, 4, 10] is a paradigm in which an algorithm alternatively makes prediction on a received data and then updates the model given the feedback of prediction to optimize a target performance metric. It has attracted tremendous attention due to its efficiency in handling large-scale and/or streaming data, and has been actively investigated for decades. While many studies are devoted to learning with traditional performance metrics (e.g., classification error rate), there has also been an increasing interest in designing efficient online learning algorithms to maximize F-measure for tackling large-scale streaming data or by one pass of large-scale batch data [3, 9, 14, 23]. This is because F-measure is more suited for imbalanced classification data because it enforces a better balance between performance on the rare class and the dominating class. Imbalanced data can be found in many applications, e.g., medical diagnostics [13], spam email detection [20], malicious URL detection [27], etc.

Online F-measure optimization is much more challenging than traditional online learning with pointwise loss functions since F-measure is non-decomposable over training examples and is a non-convex function of model parameters. Nevertheless, several previous works have made efforts to tackle this difficult problem [3, 9, 14, 23], which fall into three categories. The first category is based on minimizing a surrogate loss function or maximizing a surrogate reward function in an online fashion. This type of approaches usually has large memory costs and/or lacks statistical consistency for F-measure due to that the consistency/calibration of surrogate loss of F-measure is not clear.

---

[*]equal contribution

Table 1: Comparison with Existing Work of Online F-measure Optimization. The comparison is based on fixing the total number of processed examples to be $n$, where $d$ is the dimensionality of data, $m > 0$ is a parameter of the referred algorithm, and $\alpha > 0$. Comp. costs is short for computational costs.

| | Target of Convergence Analysis | Convergence Result | Consistency | Memory Costs | Comp. Costs |
|---|---|---|---|---|---|
| [9] | Empirical Structural Surrogate Loss | $O(1/n^{1/4})$ | No | $O(\sqrt{n}d)$ | $O(nd)$ |
| [14] | Surrogate F-measure | $O(1/\sqrt{n})$ | No | $O(d)$ | $O(nd)$ |
| [23] | Cost-sensitive loss/F-measure | $O(1/\sqrt{n})/O(1/m + 1/\sqrt{n})$ | Yes (iff $m = n^{\alpha}$) | $O(md)$ | $O(mnd)$ |
| [3] | Optimal Threshold/F-measure | asymptotic/asymptotic | Yes | $O(d)$ | $O(nd)$ |
| This work | Optimal Threshold/F-measure | $\widetilde{O}(1/\sqrt{n})$/asymptotic | Yes | $O(d)$ | $O(nd)$ |

The second category of approaches leverages the characterization that F-measure optimization is equivalent to a cost-sensitive loss minimization and uses online learning algorithms for minimizing a cost-sensitive loss. However, the optimal costs in this characterization depend on the optimal F-measure, which makes this type of approaches suffer from a large computational cost for tuning or searching the optimal costs. The third family of methods is based on a result that the optimal classifier for maximizing the F-measure can be achieved by thresholding the posterior class probability. Then, the problem reduces to learning the optimal threshold and the posterior probability incrementally. Online learning of the posterior probability can implemented by minimizing calibrated surrogate loss functions (e.g., logistic loss). However, the challenge of this type of approach lies at how to learn the optimal threshold on-the-fly.

In this paper, we address **this challenge (online learning of the optimal threshold)** in an elegant way. In particular, we cast the problem of learning the optimal threshold as stochastic strongly convex optimization. Nevertheless, the existing online gradient descent method with $\widetilde{O}(1/n)$ convergence for minimizing strongly convex functions is not directly applicable. The reason is that **the strong convexity parameter is unknown and unbiased stochastic gradient is not available**. *The significance of this work is to address these challenges by a new design of online algorithm and novel high probability analysis of the proposed algorithm.* Our main contributions are summarized below:

- We propose a Fast Online F-measure Optimization (FOFO) with a novel component for learning the optimal threshold for a probabilistic binary classifier. The proposed algorithm has low memory and computational costs.

- We prove an $\widetilde{O}(1/\sqrt{n})$ [2] convergence rate for learning the optimal threshold and the consistency of F-measure optimization at the population level under a point-wise convergence condition of learning the posterior probability. It is provably faster than its predecessor [3] that updates the threshold based on a heuristic.

- We conduct extensive experiments comparing the proposed algorithm with existing algorithms in the three categories. Experimental results show that FOFO has much better online and testing performance than other algorithms especially on highly imbalanced datasets.

## 2 Related Work

This work is motivated by addressing the deficiencies of previous algorithms of OFO and is also built on existing results for F-measure optimization. In this section, we will highlight them. As mentioned before previous OFO algorithms can be organized into three categories. In review of related work, we focus on the F-measure optimization part, though some of them also include contributions for optimizing other non-decomposable metrics (e.g., AUC, Precision) [9, 14, 23].

Two representative algorithms in the first category are proposed in [9, 14]. In particular, Kar et al. [9] developed an online gradient descent method for minimizing structural surrogate loss, which was motivated by a batch-learning method for optimizing the structural surrogate loss [7]. The authors of [9] established a convergence rate for minimizing the empirical structural surrogate loss in the order of $O(1/n^{1/4})$, where $n$ is the total number of processed examples. One deficiency of their algorithm is the large-memory costs due to that it needs to maintain $O(\sqrt{n})$ examples for computing the gradient of the structural surrogate loss for each update of the model parameters. Narasimhan et al. [14] addressed the issue of high-memory cost by optimizing a surrogate F-measure that is defined based on surrogate reward functions for approximating true-positive and true-negative rate. They leveraged

the pseudo-linear property of the (surrogate) F-measure and developed an alternate-maximization algorithms for optimizing the surrogate F-measure, which has a stochastic version called STAMP with a convergence rate of $O(1/\sqrt{n})$. However, their stochastic algorithm is not designed for online learning where online performance is important. In particular, their algorithm alternates between two stages with one stage updating the models using received examples and another stage updating the so-called challenge level using received examples. In contrast, our algorithm simultaneously updating the probabilistic classifier and its threshold for making predications, making it more suitable for online learning. Another deficiency of both works [9, 14] is the lack of statistical consistency of F-measure optimization at the population level.

A related algorithm in the second category of OFO is proposed in [23], which is based on minimizing cost-sensitive loss. It is motivated by that F-measure maximization is equivalent to a cost-sensitive error minimization that consists of a weighted sum of false positive and false negative [16]. Nevertheless, the optimal weight is dependent on the optimal F-measure and thus is not available. To tackle the unknown optimal weight, Yan et al. [23] proposed to learn multiple cost-sensitive classifiers corresponding to multiple settings of the weight. For online prediction, they also maintain and update selection probabilities of the multiple classifiers. At each iteration, the algorithm selects one classifier for making prediction and updates all classifiers upon receiving the label information of received data. As a result, their algorithm suffers from high memory and computational costs. They proved a convergence result for F-measure optimization by utilizing the fact that cost-sensitive surrogate loss is calibrated [19]. Nevertheless, the consistency of F-measure optimization requires the number of maintained classifiers (denoted by $m$ in Table 1) to be very large.

Recently, Busa-Fekete et al. [3] proposed a remarkably simple OFO algorithm, which belongs to the third category. It is based on a fact that optimal F-measure can be achieved by thresholding the true posterior probability of positive class [24, 15]. Hence, an online algorithm for updating the model of the posterior probability and updating the threshold is developed in [3]. Their update for the threshold is based on a heuristic by setting the threshold as half of the F-measure computed on the historical examples. This is motivated by fact established in [26] that the optimal threshold for the true posterior probability is half of the optimal F-measure. However, it is generally not true that for any probabilistic classifier the optimal threshold is half of its F-measure. As a result, they can only prove asymptotic convergence (with $n \to \infty$) for learning the optimal threshold even using the true posterior probability at each iteration. In contrast, we overcome this shortcoming by learning the optimal threshold through solving a strongly convex optimization problem. With careful design and analysis of the proposed algorithm, we are able to prove a convergence rate of $\widetilde{O}(1/\sqrt{n})$ for learning the optimal threshold under a mild condition for learning the posterior class probability.

We note that the consistency of F-measure optimization is an important concern for the design and analysis of OFO algorithm [15]. It requires that given infinite amount of data, the learned classifier should achieve the best F-measure at the population level. As a summary, we present a quick comparison between this work and related studies in Table 1 from various perspectives, including theoretical convergence results, consistency of F-measure optimization, memory and computational costs, where linear models are assumed for different algorithms in order to compare the memory and computational costs. Finally, we emphasize that although there are some batch-learning based F-measure optimization algorithms [15, 24], the comparison in this paper focuses on online algorithms.

## 3 Preliminaries and Notations

Let $\mathbf{z} = (\mathbf{x}, y)$ denote a random data, where $\mathbf{x} \in \mathcal{X}$ represents the feature vector and $y \in \{1, 0\}$ represents the binary class label. Let $\mathcal{Z} = \mathcal{X} \times \{1, 0\}$ denote the domain of the data. We assume $\mathbf{z}$ follows an unknown distribution $\mathcal{P}$, and denote the marginal distribution of the feature $\mathbf{x}$ by $\mu(\mathbf{x})$. We denote the probability of positive class by $\pi = \Pr(y = 1)$, and the true posterior probability of the positive class by $\eta(\mathbf{x}) = \Pr(y = 1|\mathbf{x})$, and thus we have $\pi = \int_{\mathbf{x} \in \mathcal{X}} \eta(\mathbf{x}) d\mu(\mathbf{x})$. Since we assume the received examples follow a distribution, in the sequel we use online gradient descent (OGD) and stochastic gradient descent (SGD) interchangeably.

Let $\mathcal{F} = \{f : \mathcal{X} \to \{1, 0\}\}$ be the set of all binary classifiers on $\mathcal{X}$. The F-measure (in particular $F_1$ measure) of $f$ at the population level is defined as

$$F(f) = \frac{2 \int_{\mathcal{X}} \eta(\mathbf{x}) f(\mathbf{x}) d\mu(\mathbf{x})}{\int_{\mathcal{X}} \eta(\mathbf{x}) d\mu(\mathbf{x}) + \int_{\mathcal{X}} f(\mathbf{x}) d\mu(\mathbf{x})}. \tag{1}$$

Denote by $F_* = \arg\max_{f \in \mathcal{F}} F(f)$. Let $\mathcal{G} = \{g : \mathcal{X} \to [0,1]\}$ denote a set of probabilistic classifier that assigns to any example $\mathbf{x}$ a probability that it belongs to the positive class. It induces a family of thresholded binary classifiers $\mathcal{H} = \{g_\theta(\mathbf{x}) := \mathcal{I}_{[g(\mathbf{x}) \geq \theta]}\} \subseteq \mathcal{F}$, where $\mathcal{I}$ is an indicator function, and $\theta \in [0,1]$ is a threshold.

It was shown that the optimal binary classifier that maximizes the F-measure at the population level can be achieved by thresholding the true posterior probability $\eta(\mathbf{x})$, i.e., $\eta_\theta(\mathbf{x}) = \mathcal{I}_{[\eta(\mathbf{x}) \geq \theta]}$ [24, 15]. As a result, we have $\max_{f \in \mathcal{F}} F(f) = \max_{\theta \in [0,0.5]} F(\eta_\theta)$. This reduces the problem of F-measure optimization into two sub-problems: learning the posterior probability $\eta(\mathbf{x})$ and learning the optimal threshold. The best threshold $\theta_*$ that maximizes $F(\eta_\theta)$ has a relationship with the optimal F-measure, i.e., $\theta_* = F_*/2$ [26]. It also implies that the best optimal threshold $\theta_* \in [0, 0.5]$.

**Definition 1.** *An algorithm is said to be F-measure consistent if the learned classifier $f$ satisfies $F_* - F(f) \xrightarrow{p} 0$, as $n \to \infty$, where $\xrightarrow{p}$ denotes convergence in probability.*

Let $\mathcal{W} = [0, 0.5]$ and $\mathcal{B}(\theta_0, r) = \{\theta : |\theta - \theta_0| \leq r\}$. Denote by $\Pi_{\mathcal{W}}[\theta]$ by a projection of $\theta$ into the domain $\mathcal{W}$. Denote by $\mathcal{X}_\theta = \{\mathbf{x} \in \mathcal{X} : \eta(\mathbf{x}) \geq \theta\}$ for any $\theta \in [0, 0.5]$ and by $\rho_\theta = \int_{\mathbf{x} \in \mathcal{X}_\theta} d\mu(\mathbf{x})$.

# 4  Fast Online F-measure Optimization Algorithm

From the discussion above, we can cast OFO into two sub-problems, i.e., online learning of the posterior probability and online learning of the optimal threshold. Let us first discuss the first sub-problem and then focus on the second sub-problem. For the first sub-problem, we assume that there exists an online algorithm $\mathcal{A}$ that can incrementally learn the posterior probability. To better illustrate this, let us consider a scenario that the true posterior probability is specified by a generalized linear model, i.e., with an appropriate feature mapping $\phi(\mathbf{x}) \in \mathbb{R}^d$ there exists $\mathbf{w}_* \in \mathbb{R}^d$ such that

$$\eta(\mathbf{x}) = \Pr(y = 1|\mathbf{x}) = \frac{1}{1 + \exp(-\mathbf{w}_*^\top \phi(\mathbf{x}))}. \tag{2}$$

It is not difficult to show that the model parameter $\mathbf{w}_*$ can be learned by minimizing the expected logistic loss (see supplement), i.e.,

$$\mathbf{w}_* \in \arg\min_{\mathbf{w} \in \mathbb{R}^d} L(\mathbf{w}) \triangleq \mathbb{E}_{\mathbf{x},y} \log(1 + \exp(-(2y - 1)\mathbf{w}^\top \phi(\mathbf{x}))). \tag{3}$$

Therefore, one can use existing online learning algorithms (e.g., SGD [28, 17]) to learn $\mathbf{w}_*$. To this end, we denote by

$$\mathbf{w}_t = \mathcal{A}(\mathbf{w}_{t-1}, \mathbf{x}_t, y_t), \tag{4}$$

the update of an online algorithm that updates the model parameter $\mathbf{w}_{t-1}$ iteratively, where $t = 1, \ldots, n$. In the next section, we discuss some choices of the online algorithm $\mathcal{A}$ and its implication for the convergence result. At the $t$-th iteration (before receiving the $t$-th example), let $\widehat{\eta}_t(\mathbf{x})$ denote an estimate of the posterior probability. For example of generalized linear model considered above, $\widehat{\eta}_t(\mathbf{x})$ can be computed by

$$\widehat{\eta}_t(\mathbf{x}) = 1/(1 + \exp(-\bar{\mathbf{w}}_{t-1}^\top \phi(\mathbf{x}))), \tag{5}$$

where $\bar{\mathbf{w}}_{t-1}$ is a solution computed based on $\mathbf{w}_0, \ldots, \mathbf{w}_{t-1}$ that has a convergence guarantee (please see Assumption 1 and its discussion in next section). It is notable that online learning of the posterior probability is also used in [3].

Next, we focus on online learning of the optimal threshold $\theta_*$. According to [26], $\theta_*$ is achieved at the unique root of

$$q(\theta) = \pi\theta - \mathbb{E}_{\mathbf{x}} \left[(\eta(\mathbf{x}) - \theta)_+\right],$$

where $q(\theta)$ is continuous and strictly increasing, and the function $(\cdot)_+$ is defined as $(x)_+ = \max(x, 0)$. Instead, we will cast the problem of learning the optimal threshold $\theta_*$ as the following strongly convex optimization problem.

**Lemma 1.** $\theta_*$ *is the unique optimizer of the following strongly convex function*

$$\min_{\theta \in [0,0.5]} Q(\theta) \triangleq \frac{1}{2} \mathbb{E}_{\mathbf{x}} \left[(\eta(\mathbf{x}) - \theta)_+^2\right] + \frac{1}{2}\pi\theta^2. \tag{6}$$

*Indeed, $Q(\theta)$ is a $\sigma$-strongly convex function with $\sigma = \pi + \min_{\theta \in [0,0.5]} \rho_\theta$.*

---

**Algorithm 1** FOFO$(n)$

---

1: Set $\mathbf{w}_0 = \mathbf{0}$, $\widehat{\theta}^{(0)} = 0$, $m = \lfloor \frac{1}{2} \log_2 \frac{2n}{\log_2 n} \rfloor - 1$, $n_0 = \lfloor n/m \rfloor$, $\widehat{\pi}_0 = 0$, $R_0 = 0.5$.

2: **for** $k = 1, \ldots, m$ **do**

3:      Set $\gamma_k = \frac{R_{k-1}}{\sqrt{10n_0}}$, $R_k = R_{k-1}/2$

4:      $\left[ \mathbf{w}^{(k)}, \widehat{\theta}^{(k)}, \widehat{\pi}^{(k)} \right] = \text{SFO}(\mathbf{w}^{(k-1)}, \widehat{\theta}^{(k-1)}, \widehat{\pi}^{(k-1)}, n_0, \gamma_k, R_{k-1}, (k-1)n_0)$

5: **end for**

---

**Algorithm 2** SFO$(\mathbf{w}, \theta, \widehat{\pi}, T, \gamma, R, T_0)$

---

1: Initialize $\bar{\theta}_1 = \theta_1 = \theta$, $\widehat{\pi}_{T_0} = \widehat{\pi}$, $\mathbf{w}_{T_0} = \mathbf{w}$

2: **for** $\tau = 1, \ldots, T$ **do**

3:      Let $t = \tau + T_0$ be the global iteration index.

4:      Receive an example $\mathbf{x}_t$

5:      Compute $\widehat{\eta}_t(\mathbf{x}_t)$ according to (5)

6:      **if** $\widehat{\eta}_t(\mathbf{x}_t) \geq \theta_\tau$ **then**

7:          Make a prediction $\hat{y}_t = 1$

8:      **else**

9:          Make a prediction $\hat{y}_t = 0$

10:     **end if**

11:     Observe the label $y_t$

12:     $\widehat{\pi}_t = \frac{1}{t}((t-1)\widehat{\pi}_{t-1} + \mathbb{I}_{[y_t=1]})$

13:     $\theta_{\tau+1} = \Pi_{\mathcal{W} \cap \mathcal{B}(\theta_1, R)}(\theta_\tau - \gamma \partial \widehat{Q}_t(\theta_\tau))$

14:     $\bar{\theta}_{\tau+1} = \frac{1}{\tau+1}(\tau \bar{\theta}_\tau + \theta_{\tau+1})$

15:     $\mathbf{w}_t = \mathcal{A}(\mathbf{w}_{t-1}, \mathbf{x}_t, y_t)$

16: **end for**

17: **return** $\mathbf{w}_t, \bar{\theta}_{T+1}, \widehat{\pi}_t$

---

The above lemma can be easily shown by noting that $\nabla Q(\theta) = q(\theta)$. The strong convexity of $Q(\theta)$ is also not difficult to prove (see supplement for details). The next lemma shows that if $\theta$ is closer to $\theta_*$ then $F(\eta_\theta)$ is closer to $F_*$, which further justifies our approach of learning the optimal threshold by optimizing the strongly convex function $Q(\theta)$.

**Lemma 2.** $\forall \theta \in [0, 0.5]$, *there exists* $c > 0$ *such that* $F_* - F(\eta_\theta) \leq c|\theta - \theta_*|$.

The proof of this lemma can be found in the supplement. However, there are several challenges for minimizing $Q(\theta)$ to have a faster convergence. First, even an unbiased stochastic gradient of $Q(\theta)$ is not available due to that $\eta(\mathbf{x})$ is not given aprior. Only a noisy estimation of $\widehat{\eta}_t(\mathbf{x})$ is available through the historical model parameters. Second, the strongly convex parameter $\sigma \geq \pi$ of $Q(\theta)$ is unknown. Standard SGD method for minimizing a $\sigma$-strongly convex function with an $\widetilde{O}(1/(\sigma n))$ convergence rate on the objective value requires knowing the strong convexity parameter for setting the step size. One may use historical examples $y_1, \ldots, y_{t-1}$ to obtain a new estimate of $\pi$ at each iteration $t$ and use it to set the step size. However, analysis of such an approach is difficult. Another simple approach is to use a dedicated set of examples to obtain a lower bound of $\pi$ and then use it to set the step size. Nevertheless, such an approach could yield a large convergence error because the lower bound of $\pi$ could be very small especially when the data is highly imbalanced.

We address these issues by proposing a novel stochastic algorithm that does not require using the strong convexity parameter to set the step size, and also can tolerate moderate noise in $\widehat{\eta}_t(\mathbf{x})$ to enjoy a fast convergence rate of $\widetilde{O}(1/(\sigma n))$ for minimizing $Q(\theta)$. We present the Fast Online F-measure Optimization (FOFO) in Algorithm 1 and Algorithm 2, where $\widehat{\pi}_t$ in Step 12 is the estimate of $\pi$ up to the $t$-th iteration, i.e., $\widehat{\pi}_t = \sum_{\tau=1}^{t} y_\tau / t$ and $\widehat{Q}_t(\theta) = \frac{1}{2}(\widehat{\eta}_t(\mathbf{x}_t) - \theta)_+^2 + \frac{1}{2}\widehat{\pi}_t \theta^2$. It is worth mentioning that the Step 5 and 15 in Algorithm 2 can be replaced by other online learners of the posterior probability which are not necessarily restricted to the generalized linear model (2). The main algorithm FOFO is presented in a way that facilitates the analysis. The updates of FOFO are divided into $m$ stages, where at each stage a stochastic F-measure optimization method (Algorithm 2) is called for running $n_0$ iterations. For each received example $\mathbf{x}_t$, the prediction $\hat{y}_t$ is computed by thresholding current estimate of posterior probability $\widehat{\eta}_t(\mathbf{x}_t)$ by the current value of $\theta$. At each iteration of a

stage $k$, we use the gradient of $\widehat{Q}_t(\theta)$ to update $\theta$ with a constant step size $\gamma_k$ and project it into $\mathcal{W} \cap \mathcal{B}(\theta_1, R_{k-1})$, where $\theta_1$ is the initial solution of this stage and $R_{k-1}$ is a radius parameter. The step-size $\gamma_k$ and radius parameter $R_{k-1}$ are changed according to Step 3 in Algorithm 1. We remark that the same multi-stage scheme (especially the setting of $m$ and $n_0$) Algorithm 1 is due to [8], which was also used in several stochastic optimization algorithms for solving different problems [11, 12]. However, the difference from these studies is that $\partial \widehat{Q}_t(\theta)$ is not an unbiased stochastic gradient of $Q(\theta)$.

## 5 Convergence and Consistency Results of FOFO

In this section, we further justify the proposed FOFO by presenting a convergence result of FOFO for learning the optimal threshold $\theta_*$ and a consistency result of FOFO for F-measure optimization. Omitted proofs can be found in the supplement.

For simplicity of analysis, we let $\phi(\mathbf{x}) = \mathbf{x}$ and w.l.o.g assume that feature vectors are bounded by a positive number $\kappa$, i.e., $\sup_{\mathbf{x} \in \mathcal{X}} \|\mathbf{x}\|_2 \le \kappa$. As mentioned in the last section that $\partial \widehat{Q}_t(\theta)$ is not an unbiased stochastic gradient of $Q(\theta)$. Nevertheless, we expect that $\partial \widehat{Q}_t(\theta)$ is getting close to an unbiased stochastic gradient of $Q(\theta)$ as $\widehat{\eta}_t(\mathbf{x})$ converges to $\eta(\mathbf{x})$. To formalize this notion, we will introduce the following assumption about convergence of the model for the posterior probability.

**Assumption 1.** *Assume there exists an online algorithm $\mathcal{A}$ that learns the posterior probability at a rate of $1/\sqrt{t}$, i.e., by learning with $t$ examples the error of learned posterior probability $\widehat{\eta}_t(\cdot)$ is $\max_{\mathbf{x} \in \mathcal{X}} |\widehat{\eta}_t(\mathbf{x}) - \eta(\mathbf{x})| \le O(1/\sqrt{t})$ with high probability.*

**Remark:** Please note that this is a high-level assumption without imposing any form of the posterior probability. The assumed $O(1/\sqrt{t})$ rate for learning the posterior probability is the minimal assumption for achieving an $O(1/\sqrt{n})$ rate of the learning the threshold. It is worth mentioning that estimating the posterior probability can be considered as a special problem of statistical density estimation, which has been studied in the literature with a convergence rate as fast as $O(1/t)$ [25, 6, 21]. We provide a justification here for the considered generalized linear model (2), which is Lipchitz continuous with respect to $\mathbf{w}$. To this end, it suffices to assume that there exists an algorithm $\mathcal{A}$ as in (4) that produces solutions $\bar{\mathbf{w}}_t$ converging to $\mathbf{w}_*$ at a rate of $O(1/\sqrt{t})$ with high probability, i.e., we have $\|\bar{\mathbf{w}}_t - \mathbf{w}_*\|_2 \le \frac{C'}{\sqrt{t}}$ with high probability, where $\bar{\mathbf{w}}_t$ is a solution computed from $\mathbf{w}_1, \dots, \mathbf{w}_t$ and $C'$ is a problem-dependent value. This can be justified as following. Note that the objective function $L(\mathbf{w})$ in (3) is strongly convex for $\mathbf{w}$ in a compact domain if the data covariance matrix is nonsingular [1]. Thus SGD method for strongly convex function using a suffix-averaging solution $\bar{\mathbf{w}}_t = (\mathbf{w}_{(1-\alpha)t+1} + \dots + \mathbf{w}_t)/(\alpha t)$ with $\alpha \in (0,1)$ can have an $O(\log(1/\delta)/t)$ convergence rate for minimizing $L(\mathbf{w})$, i.e., $L(\bar{\mathbf{w}}_t) - L(\mathbf{w}_*) \le O(\log(1/\delta)/t)$ [17], which implies $\|\bar{\mathbf{w}}_t - \mathbf{w}_*\|_2 \le \frac{C'}{\sqrt{t}}$ with a high probability $1 - \delta$ for $\delta \in (0,1)$ and $C' = O(\sqrt{\log(1/\delta)})$. When the convariance matrix is not singular, by Corollary 7 of [22], a quadratic growth condition is satisfied and we can still get the $O(1/t)$ convergence for minimizing $L(\mathbf{w})$ via accelerated stochastic subgradient method proposed in [22]. Even though the strong convexity parameter of $L(\mathbf{w})$ is not exploited, the stochastic approximation algorithm proposed in [2] with a constant step size also has an $O(1/t)$ convergence rate of an averaged solution $\bar{\mathbf{w}}_t$ for minimizing $L(\mathbf{w})$ with a large probability (e.g., 0.99). These results would limply an $O(1/\sqrt{t})$ convergence for $\|\bar{\mathbf{w}}_t - \mathbf{w}_*\|$ for an optimal solution $\mathbf{w}_* \in \arg\min_{\mathbf{w} \in \mathcal{W}} L(\mathbf{w})$. In the following results, we will assume that $\max_{\mathbf{x} \in \mathcal{X}} |\widehat{\eta}_t(\mathbf{x}) - \eta(\mathbf{x})| \le O(\sqrt{\log(1/\delta)/t})$ holds with a high probability $1 - \delta$. The results can be extended to the case that it holds with a large constant probability.

We first state the convergence result of one stage of FOFO, i.e., SFO.

**Theorem 2 (Convergence Result of SFO).** *Suppose Assumption 1 holds with high probability $1 - \delta$ in the sense that $\max_{\mathbf{x} \in \mathcal{X}} |\widehat{\eta}_t(\mathbf{x}) - \eta(\mathbf{x})| \le C\kappa\sqrt{\log(1/\delta)}/\sqrt{t}$. If $|\theta_1 - \theta_*| \le R$, running Algorithm 2 for $T$-iterations with $\gamma = \frac{R}{\sqrt{10T}}$, we have with probability at least $1 - \delta$,*

$$Q(\bar{\theta}_T) - Q(\theta_*) \le \frac{2\sqrt{10}R + R(20 + 4C\kappa)\sqrt{\ln(12T/\delta)}}{\sqrt{T}}.$$

**Remark:** The above result indicates the FOFO has at least an $O(1/\sqrt{n})$ convergence for optimizing $Q(\theta)$. The next theorem establishes a faster convergence $\widetilde{O}(1/n)$ by utilizing the above result and the multi-stage scheme of FOFO.

**Theorem 3** (**Convergence Result of FOFO**). *Given $\delta \in (0, 1)$, under the same condition in Theorem 2 and $n$ is sufficiently large such that $n > 100$. Then with probability at least $1 - \delta$,*

$$Q(\widehat{\theta}_m) - Q(\theta_*) \leq \widetilde{O}\left(\frac{\log(\frac{1}{\delta})}{\sigma n}\right).$$

**Remark:** Since $Q(\theta)$ is $\sigma$-strongly convex, the above result implies that $|\widehat{\theta}_m - \theta_*| \leq \widetilde{O}(1/(\sigma\sqrt{n}))$.

Finally, by the convergence of $\widehat{\theta}_m$ and $\widehat{\eta}_n(\mathbf{x})$, we can establish the consistency of FOFO for F-measure optimization by using Proposition 13 in [15].

**Theorem 4** (**Consistency of F-measure Optimization**). *Suppose Assumption 1 holds, then FOFO is F-measure consistent, i.e., the final binary classifier $[\widehat{\eta}_n(\mathbf{x}) \geq \widehat{\theta}_m]$ is F-measure consistent.*

We would like to mention that for establishing the consistency of FOFO for F-measure optimization a weaker assumption about the convergence of $\widehat{\eta}_t$ can be used. As long as $\max_{\mathbf{x} \in \mathcal{X}} |\widehat{\eta}_t(\mathbf{x}) - \eta(\mathbf{x})| \leq O(1/t^\alpha)$ for some $\alpha > 0$, a convergence result can be established for $\widehat{\theta}_m$, which implies the consistency of FOFO for F-measure optimization by the Proposition 13 in [15].

**Extension to Other Metrics.** Before ending this section, we would like to mention that the proposed method can be extended to other non-decomposable metrics such as Jaccard similarity coefficient (JAC) and $F_\beta$ measure. We present more details in the supplement for interested readers.

## 6   Experiments

We present some experimental results in this section. We will compare with four baselines including online learning with logistic loss using $0.5$ for thresholding the posterior probability (referred as LR), STAMP [14], OMCSL [23], and OFO [3]. The last three are representative algorithms from the three categories. For LR, OFO and FOFO, we use the same SGD for learning the posterior probability by minimizing the logistic loss. To be fair, we implement the STAMP algorithm with a logistic loss based reward function, and also use logistic loss for OMCSL.

We evaluate the performance on 25 binary classification tasks from seven benchmark datasets (covtype, webspam, a9a, ijcnn1, w8a, sensorless, protein). All the datasets involved are downloaded from the LIBSVM repository [5]. It is notable that covtype, sensorless and protein are multi-class datasets. We construct binary tasks following the scheme one vs others denoted by "X vs o" below. Each dataset is divided into three parts (1:1:1) for online training, online validation and offline testing. The validation data is used to select the best parameters by running the considered algorithms and selecting the best parameters according to the final F-measure. In particular, for FOFO, OFO, and LR, we tune the initial step parameter for learning the posterior probability in the range $2^{[-4:1:4]}$. For STAMP and OMCSL, the stepsize parameter is also tuned in $2^{[-8:1:4]}$. For OMCSL, we use 10 settings for the weights and learn 10 classifiers online. For each data, we repeat the experiments 10 times by running on 10 random shuffled data and report the average and variance. We will report online performance (evaluation of predictions on historical examples) on online training data, and offline performance by evaluating the final models on the testing data.

Due to limitation of space, we only report part of the results (complete results are included in the supplement). The online performance (F-score vs iterations) is plotted in Figure 1 and 2 for covtype datasets and other datasets. The F-measure vs running time s plotted in Figure 3 and 4 and the offline testing performance are reported in the supplement. We can first consider the results on covtype. From (a) to (f) in Figure 1 where $p = x\%$ denotes the percentage of positive examples, we organize the data in the order of increasing imbalance. We can see that as data becomes more imbalanced, the improvement of our algorithm FOFO over baselines becomes larger. On the datasets that are more balanced (covtype 2 vs o, 1 vs o), the difference between FOFO and OFO is small, and they both outperforms other baselines. When the datasets become highly imbalanced (covtype 6 vs o, 5 vs o), the baseline LR becomes extremely worse, and the margin of FOFO over OFO becomes larger. The comparison between FOFO and OFO verifies that the proposed method for learning the optimal threshold converges faster as they share the same component for learning the posterior probability. The comparison between FOFO and other baselines verify that the proposed method is better than

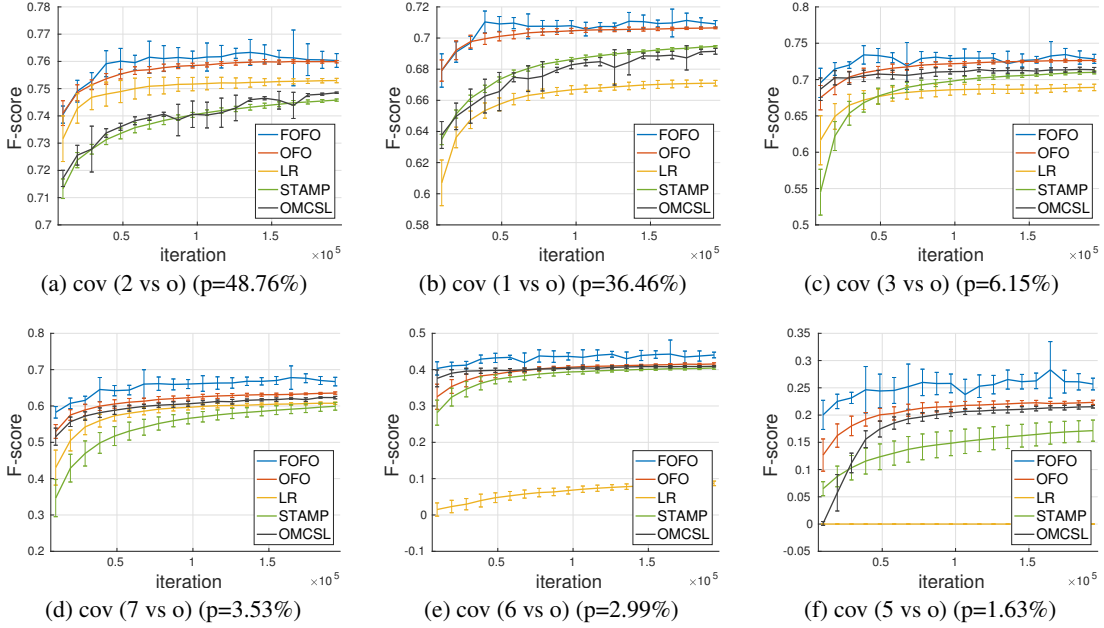

Figure 1: Online Performance of F-measure for covtype dataset

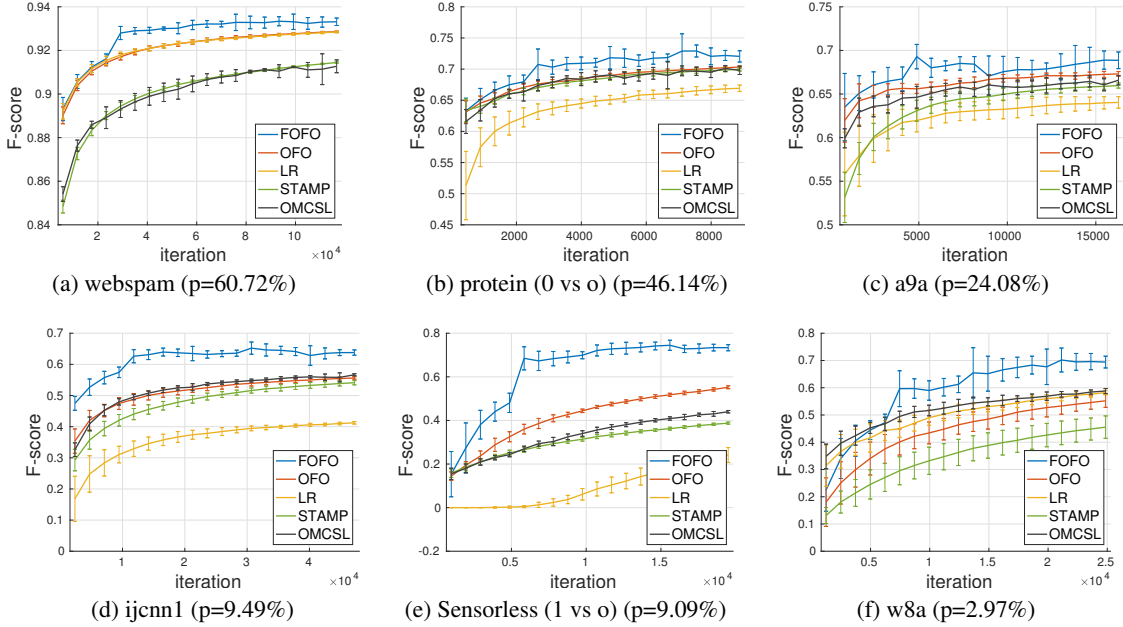

Figure 2: Online Performance of F-measure for other datasets

other categories of algorithms. The results on other datasets in Figure 2 also demonstrate that FOFO is faster than OFO and other baselines especially for highly imbalanced data. The running time results in Figure 3 and 4 also verify that the proposed FOFO algorithm is the most efficient.

## 7 Conclusions

In this paper, we proposed a fast online F-measure optimization algorithm with low memory and computational costs by learning the optimal threshold for a probabilistic classifier. A novel stochastic algorithm was proposed for learning the optimal threshold. We prove that the proposed algorithm for learning of the optimal threshold has a convergence rate $\widetilde{O}(1/\sqrt{n})$, and the proposed algorithm enjoys F-measure consistency at the population level. Extensive experimental results comparing

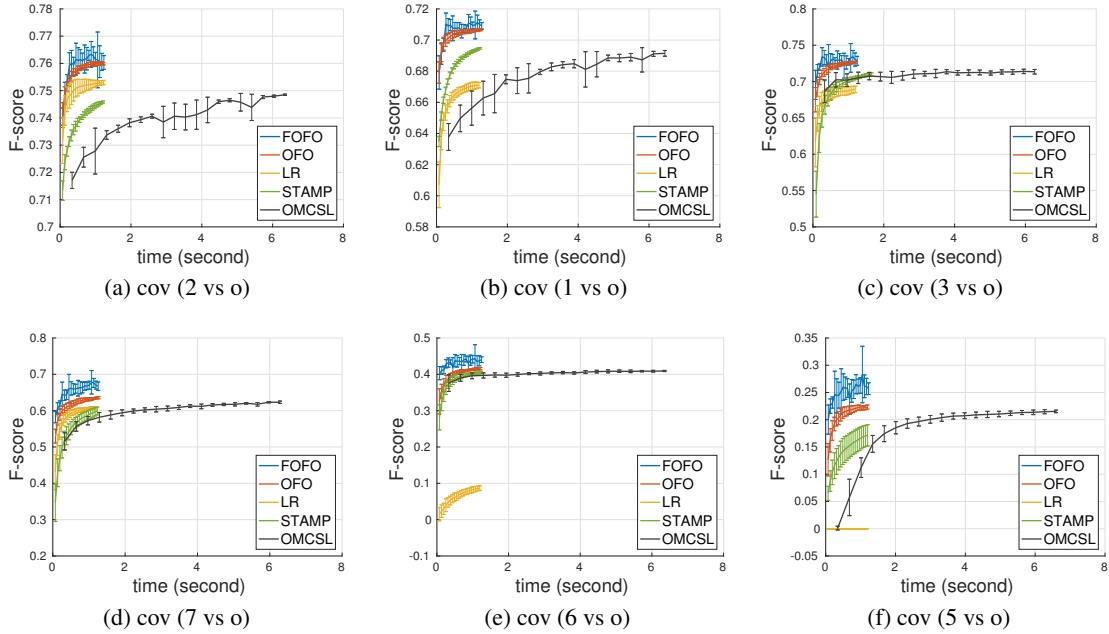

Figure 3: Online F-measure vs Running Time for covtype dataset

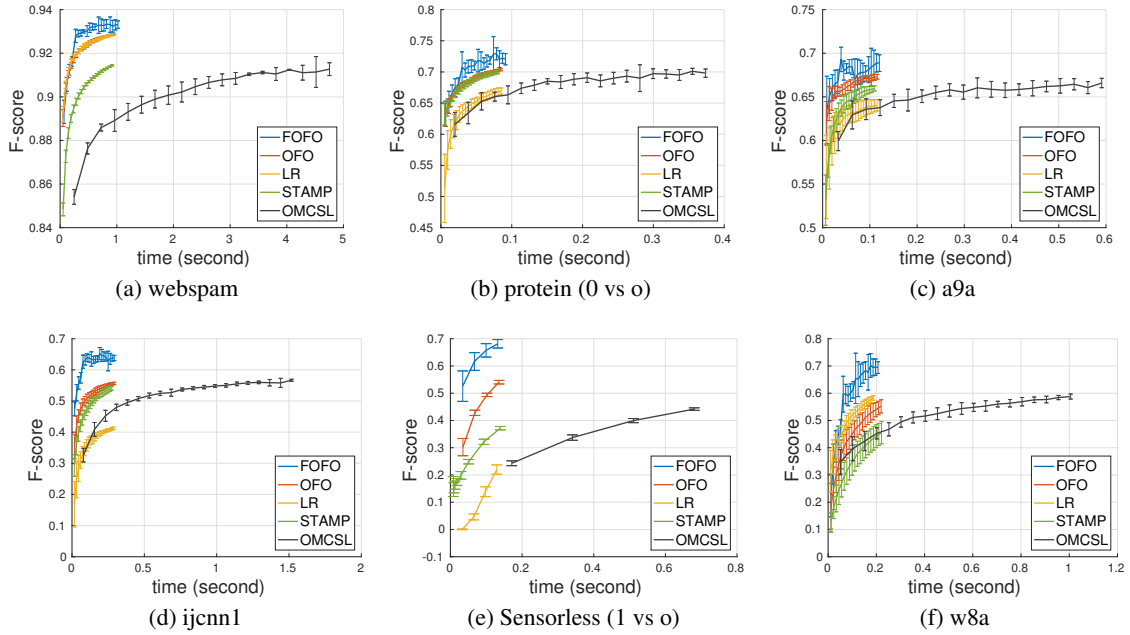

Figure 4: Online F-measure vs Running Time for other datasets

with state-of-the-art online F-measure optimization algorithms also demonstrate the efficiency of the proposed algorithm, especially on highly imbalanced datasets.

## Acknowledgement

The authors thank the anonymous reviewers for their helpful comments. M. Liu, X. Zhang and T. Yang are partially supported by National Science Foundation (IIS-1545995).

## Footnotes

[2]The $\widetilde{O}(\cdot)$ notation hides logarithmic factors. This convergence rate is implied by a convergence rate of $\widetilde{O}(1/n)$ for minimizing the involved strongly convex function.

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
