[Supplementary Material]

# Supplementary Material for "Faster Online Learning of Optimal Threshold for Consistent F-measure Optimization"

**Mingrui Liu**[*†], **Xiaoxuan Zhang**[*†], **Xun Zhou**[‡], **Tianbao Yang**[†]

[†]Department of Computer Science, The University of Iowa, Iowa City, IA 52242, USA
[‡]Department of Management Sciences, The University of Iowa, Iowa City, IA 52242, USA
`mingrui-liu, xiaoxuan-zhang, xun-zhou, tianbao-yang@uiowa.edu`

## 1 Extension to Other Metrics

In this section, we consider the extension of the proposed method to other metrics, in particular Jaccard similarity coefficient, and $F_\beta$ measure.

Let us first consider Jaccard similarity coefficient (JAC) [2]:

$$\text{JAC}(f) = \frac{\int_{\mathcal{X}} \eta(\mathbf{x}) f(\mathbf{x}) d\mu(\mathbf{x})}{\pi + \int_{\mathcal{X}} f(\mathbf{x}) d\mu(\mathbf{x}) - \int_{\mathcal{X}} \eta(\mathbf{x}) f(\mathbf{x}) d\mu(\mathbf{x})}.$$

Then we have

$$\frac{1}{\text{JAC}(f)} = \frac{\pi + \int_{\mathcal{X}} f(\mathbf{x}) d\mu(\mathbf{x})}{\int_{\mathcal{X}} \eta(\mathbf{x}) f(\mathbf{x}) d\mu(\mathbf{x})} - 1 = \frac{2}{F(f)} - 1.$$

Therefore

$$\text{JAC}(f) = \frac{F(f)}{2 - F(f)}.$$

If $F(f)$ is maximized so is $\text{JAC}(f)$. According to [2], the optimal threshold $\theta_{\text{JAC},*}$ for maximizing $\text{JCA}(\eta_\theta(\mathbf{x}))$ is given by $\theta_{\text{JAC},*} = \frac{\text{JAC}_*}{1+\text{JAC}_*} = F_*/2 = \theta_{F,*}$, where $\theta_{F,*}$ is the optimal threshold for F-measure maximization. Given an estimate of $\theta_F$ for F-measure optimization, we can set the threshold for JAC maximization as $\theta_{\text{JAC}} = \theta_F$, and when $\theta_F \to \theta_{F,*}$, we have $\theta_{\text{JAC}} \to \theta_{\text{JAC},*}$. As a result, the proposed algorithm FOFO is still applicable.

Next, let us consider $F_\beta$-measure:

$$F_\beta(f) = \frac{(1+\beta^2) \int_{\mathcal{X}} \eta(\mathbf{x}) f(\mathbf{x}) d\mu(\mathbf{x})}{\beta^2 \pi + \int_{\mathcal{X}} f(\mathbf{x}) d\mu(\mathbf{x})}.$$

Following the same analysis as in [3][Lemma 13, Lemma 14], we can have that $F_\beta(\eta_\theta)$ is maximized at a point $\theta_{\beta,*}$ that is the root of the following equation:

$$\pi \beta^2 \theta - \mathbb{E}_{\mathbf{x}} [(\eta(\mathbf{x}) - \theta)_+] = 0,$$

which is the optimal solution of the following strongly convex function

$$Q(\theta) \triangleq \frac{1}{2} \mathbb{E}_{\mathbf{x}} [(\eta(\mathbf{x}) - \theta)_+^2] + \frac{1}{2} \pi \beta^2 \theta^2.$$

---

[*]equal contribution

For the optimal threshold $\theta_{\beta,*}$ and optimal $F_{\beta,*}$, we have $\theta_{\beta,*} = \frac{F_{\beta,*}}{1+\beta^2}$. Hence, we can search for $\theta_{\beta,*}$ by solving the following problem:

$$\min_{\theta \in [0, 1/(1+\beta^2)]} Q(\theta) \triangleq \frac{1}{2}\mathbb{E}_{\mathbf{x}}\left[(\eta(\mathbf{x}) - \theta)^2_+\right] + \frac{1}{2}\pi\beta\theta^2.$$

We can modify FOFO a little to account for this change.

# 2 Missing Proofs

## 2.1 $\mathbf{w}_*$ minimizes the expected logistic loss

Under the assumption that

$$\eta(\mathbf{x}) = \Pr(y = 1|\mathbf{x}, \mathbf{w}_*) = \frac{1}{1 + \exp(-\mathbf{w}_*^\top \phi(\mathbf{x}))},$$

we prove that $\mathbf{w}_*$ is the minimizer of the following problem:

$$\min_{\mathbf{w} \in \mathbb{R}^d} L(\mathbf{w}) \triangleq \mathbb{E}_{\mathbf{x},y} \log(1 + \exp(-(2y - 1)\mathbf{w}^\top \phi(\mathbf{x}))). \tag{7}$$

Using variable change $\tilde{y} = 2y - 1$, $\Pr(\tilde{y}|\mathbf{x}, \mathbf{w}_*) = \frac{1}{1+\exp(-\tilde{y}\mathbf{w}_*^\top \phi(\mathbf{x}))}$, and $L(\mathbf{w}) = \mathbb{E}_{\mathbf{x},\tilde{y}} \log(1 + \exp(-\tilde{y}\mathbf{w}^\top \phi(\mathbf{x})))$. Then,

$$L(\mathbf{w}) = \mathbb{E}_{\mathbf{x},\tilde{y}} \log(1 + \exp(-\tilde{y}\mathbf{w}^\top \phi(\mathbf{x}))) = -\int_{\mathbf{x}} \mathbb{E}_{\tilde{y}|\mathbf{x}}[\log \Pr(\tilde{y}|\mathbf{x}, \mathbf{w})]d\mu(\mathbf{x})$$

$$= \int_{\mathbf{x}}\left[-\sum_{\tilde{y}} \Pr(\tilde{y}|\mathbf{x}, \mathbf{w}_*) \log \Pr(\tilde{y}|\mathbf{x}, \mathbf{w})\right] d\mu(\mathbf{x})$$

Note that the term in the square brackets is the KL divergence between two distributions $\Pr(\tilde{y}|\mathbf{x}, \mathbf{w}_*)$ and $\Pr(\tilde{y}|\mathbf{x}, \mathbf{w})$ plus a constant independent of $\mathbf{w}$. Therefore $\mathbf{w} = \mathbf{w}_*$ minimizes this term and hence minimizes $L(\mathbf{w})$.

## 2.2 Proof of Lemma 1

We prove the strong convexity parameter here.

$$Q(\theta) = \frac{1}{2}\int_{\eta(\mathbf{x})\geq\theta}(\theta^2 - 2\eta(\mathbf{x})\theta + \eta(\mathbf{x})^2)d\mu(\mathbf{x}) + \frac{1}{2}\pi\theta^2$$

$$= \frac{1}{2}\theta^2(\rho_\theta + \pi) - \theta \int_{\eta(\mathbf{x})\geq\theta} \eta(\mathbf{x})d\mu(\mathbf{x}) + c$$

where $\rho_\theta = \int_{\eta(\mathbf{x})\geq\theta} d\mu(\mathbf{x})$, $c$ is a constant independent of $\theta$. Then we can see the strong convexity parameter of $Q(\theta)$ over $[0, 0.5]$ is $\pi + \min_{\theta \in [0,0.5]} \rho_\theta$.

## 2.3 Proof of Lemma 2

*Proof.* For $A \subseteq \mathcal{X}$, define $\rho(A) = \int_{\mathbf{x}\in\mathcal{A}} 1 \cdot d\mu(\mathbf{x}) = \Pr(\mathbf{x} \in A)$. Let $\mathcal{X}_* = \{\mathbf{x} \in \mathcal{X}|\eta(\mathbf{x}) \geq \theta_*\}$ and $\mathcal{X}' = \{\mathbf{x} \in \mathcal{X}|\eta(\mathbf{x}) \geq \theta\}$, and note that $\eta_\theta(\mathbf{x}) = \mathbb{I}(\eta(\mathbf{x}) \geq \theta)$, we have

$$\frac{1}{2}F(\eta_\theta) = \frac{\int_{\mathcal{X}'} \eta(\mathbf{x})d\mu(\mathbf{x})}{\pi + \rho(\mathcal{X}')} \tag{8}$$

According to [3], $F(\eta_{\theta_*}) = 2\theta_*$. Thus,

$$\theta_* = \frac{1}{2}F(\eta_{\theta_*}) = \frac{\int_{\mathcal{X}_*} \eta(\mathbf{x})d\mu(\mathbf{x})}{\pi + \rho(\mathcal{X}_*)}$$

$$\int_{\mathcal{X}_*} \eta(\mathbf{x})d\mu(\mathbf{x}) = \theta_*(\pi + \rho(\mathcal{X}_*)) \tag{9}$$

Then we consider two cases based on the relation between $\theta$ and $\theta_*$.

**Case 1.** $0 \le \theta \le \theta_*$

Since $\mathcal{X}_* \subseteq \mathcal{X}'$, let $A = \mathcal{X}' - \mathcal{X}_* = \{\mathbf{x} \in \mathcal{X} | \theta \le \eta(\mathbf{x}) < \theta_*\}$. From (8),

$$\frac{1}{2}F(\eta_\theta) = \frac{\int_{\mathcal{X}_*} \eta(\mathbf{x})d\mu(\mathbf{x}) + \int_A \eta(\mathbf{x})d\mu(\mathbf{x})}{\pi + \rho(\mathcal{X}_*) + \rho(A)}$$

On $A$, we have $\eta(\mathbf{x}) \ge \theta$, thus $\int_A \eta(\mathbf{x})d\mu(\mathbf{x}) \ge \theta\rho(A)$. From (9), we have

$$\frac{1}{2}F(\eta_\theta) \ge \frac{\theta_*(\pi + \rho(\mathcal{X}_*)) + \theta\rho(A)}{\pi + \rho(\mathcal{X}_*) + \rho(A)} = \frac{\theta_*(\pi + \rho(\mathcal{X}_*)) + \theta_*\rho(A) - \theta_*\rho(A) + \theta\rho(A)}{\pi + \rho(\mathcal{X}_*) + \rho(A)}$$

$$=\theta_* - \frac{(\theta_* - \theta)\rho(A)}{\pi + \rho(\mathcal{X}_*) + \rho(A)} \ge \theta_* - (\theta_* - \theta) = \theta$$

Thus $F(\eta_{\theta_*}) - F(\eta_\theta) \le 2(\theta_* - \theta) \le \frac{2}{\pi}|\theta_* - \theta|$.

**Case 2.** $\theta_* < \theta \le 0.5$

Since $\mathcal{X}' \subseteq \mathcal{X}_*$, let $A = \mathcal{X}_* - \mathcal{X}' = \{\mathbf{x} \in \mathcal{X} | \theta_* \le \eta(\mathbf{x}) < \theta\}$. From (8),

$$\frac{1}{2}F(\eta_\theta) = \frac{\int_{\mathcal{X}_*} \eta(\mathbf{x})d\mu(\mathbf{x}) - \int_A \eta(\mathbf{x})d\mu(\mathbf{x})}{\pi + \rho(\mathcal{X}_*) - \rho(A)}$$

On $A$, we have $\eta(\mathbf{x}) < \theta$, thus $\int_A \eta(\mathbf{x})d\mu(\mathbf{x}) \le \theta\rho(A)$. From (9), we have

$$\frac{1}{2}F(\eta_\theta) \ge \frac{\theta_*(\pi + \rho(\mathcal{X}_*)) - \theta\rho(A)}{\pi + \rho(\mathcal{X}_*) - \rho(A)} = \frac{\theta_*(\pi + \rho(\mathcal{X}_*)) - \theta_*\rho(A) + \theta_*\rho(A) - \theta\rho(A)}{\pi + \rho(\mathcal{X}_*) - \rho(A)}$$

$$=\theta_* - \frac{(\theta - \theta_*)\rho(A)}{\pi + \rho(\mathcal{X}_*) - \rho(A)} \ge \theta_* - \frac{1}{\pi}(\theta - \theta_*).$$

The last step holds because $\rho(A) < 1$ and $\rho(\mathcal{X}_*) - \rho(A) = \rho(\mathcal{X}') \ge 0$. Then we have $F(\eta_{\theta_*}) - F(\eta_\theta) \le 2\left(\theta_* - \theta_* + \frac{1}{\pi}(\theta - \theta_*)\right) = \frac{2}{\pi}(\theta - \theta_*) = \frac{2}{\pi}|\theta_* - \theta|$.

We combine both cases and get the final result. $\qquad\square$

## 2.4 Proof of Theorem 2

*Proof.* Here we consider any stage $k$. Let $\tau$ denote the iteration index of SFO and $t = T_0 + \tau$ denote the global index. Define $g(\theta) = q(\theta) = \partial Q(\theta)$, $\mathbf{z} = (\mathbf{x}, y)$, $G(\theta, \mathbf{z}) = \pi\theta - (\eta(\mathbf{x}) - \theta)_+$, $\hat{G}_t(\theta, \mathbf{z}) = \hat{\pi}_t\theta - (\hat{\eta}_t(\mathbf{x}) - \theta)_+$. It is clear that $\mathbb{E}[G(\theta, \mathbf{z})] = g(\theta)$, and $\max\left(|g(\theta_\tau)|, |G(\theta_\tau, \mathbf{z}_t)|, |\hat{G}_t(\theta_\tau, \mathbf{z}_t)|\right) \le 2$ for any $\tau$. Following standard analysis of gradient descent, we have

$$\frac{1}{T}\sum_{\tau=1}^{T}(\theta_\tau - \theta_*)\hat{G}_t(\theta_\tau, \mathbf{z}_t) \le \frac{|\theta_1 - \theta_*|^2}{2\gamma T} + \frac{\gamma\max(\hat{G}_t(\theta_\tau, \mathbf{z}_t))^2}{2}$$

Then by the convexity of $Q(\theta)$, we have

$$Q(\bar{\theta}_T) - Q(\theta_*) \le \frac{\|\theta_1 - \theta_*\|_2^2}{2\gamma T} + \frac{4\gamma}{2} + \frac{\sum_{\tau=1}^{T}(\theta_\tau - \theta_*)(g(\theta_\tau) - G(\theta_\tau, \mathbf{z}_t))}{T}$$

$$+ \frac{\sum_{\tau=1}^{T}(\theta_\tau - \theta_*)(G(\theta_\tau, \mathbf{z}_t) - \hat{G}_t(\theta_\tau, \mathbf{z}_t))}{T}$$

$$=\mathbf{I} + \mathbf{II} + \mathbf{III} + \mathbf{IV}$$

Now we try to bound the four terms respectively. Note that $\mathbf{I} \le \frac{R^2}{2\gamma T}$, $\mathbf{II} \le 2\gamma$. To bound the third term, we utilize the similar analysis of SGD (e.g. [4]). Define

$$\tilde{\theta}_1 = \theta_1 \in [0, 0.5] \cap \mathcal{B}(\theta_1, R),$$

$$\tilde{\theta}_{\tau+1} = \Pi_{[0,0.5]\cap\mathcal{B}(\theta_1,R)}(\tilde{\theta}_\tau - \gamma(g(\theta_\tau) - G(\theta_\tau, \mathbf{z}_t))).$$

Then we have

$$\sum_{\tau=1}^{T}\gamma(\tilde{\theta}_{\tau}-\theta_{*})(g(\theta_{\tau})-G(\theta_{\tau},\mathbf{z}_t)) \leq \frac{\|\tilde{\theta}_1-\theta_*\|_2^2}{2} + \frac{1}{2}\sum_{\tau=1}^{T}\gamma^2\|g(\theta_{\tau})-G(\theta_t,\mathbf{z}_t)\|_2^2$$

$$\leq \frac{R^2}{2} + 8\gamma^2 T. \tag{10}$$

Note that both $\theta_{\tau}$ and $\tilde{\theta}_{\tau}$ are measurable with respect to $\mathcal{F}_{t-1}=\{\mathbf{z}_1,\ldots,\mathbf{z}_{t-1}\}$, and $\{S_{\tau}: \gamma(\theta_{\tau}-\tilde{\theta}_{\tau})(g(\theta_{\tau})-G(\theta_{\tau},\mathbf{z}_t)), \tau=1,\ldots,T\}$ is a martingale difference sequence, and for any $\tau$ we have $|\gamma(\theta_{\tau}-\tilde{\theta}_{\tau})(g(\theta_{\tau})-G(\theta_{\tau},\mathbf{z}_t))| \leq 4\gamma\|\theta_{\tau}-\tilde{\theta}_{\tau}\|_2 \leq 4\gamma \times 2R = 8\gamma R$. Then by Azuma-Hoeffding's inequality, we have with probability at least $1-\frac{\delta}{3}$,

$$\sum_{\tau=1}^{T}\gamma(\theta_{\tau}-\tilde{\theta}_{\tau})(g(\theta_{\tau})-G(\theta_{\tau},\mathbf{z}_t)) \leq 8\gamma R\sqrt{2T\ln(\frac{3}{\delta})}. \tag{11}$$

Adding (10) and (11) together suffices to show that with probability at least $1-\frac{\delta}{3}$, we have

$$\mathbf{III} \leq \frac{R^2}{2\gamma T} + 8\gamma + \frac{8R\sqrt{2\ln(\frac{3}{\delta})}}{\sqrt{T}}.$$

Next we bound $\mathbf{IV}$ according to the Lemma 3 introduced later. By union bound, we have with probability at least $1-\frac{\delta}{3}$, we have

$$\mathbf{IV} \leq \frac{1}{T}\sum_{\tau=1}^{T}\left(\sup_{\tau}(\|\theta_{\tau}-\theta_1\|_2+\|\theta_1-\theta_*\|_2)\cdot \sup_{\theta\in[0,0.5],\mathbf{z}\in\mathcal{Z}}\|\widehat{G}_t(\theta,\mathbf{z})-G(\theta,\mathbf{z})\|_2\right)$$

$$\leq \frac{2R\cdot(1+C\kappa)\times\sum_{t=1}^{T}\sqrt{\frac{\ln(12T/\delta)}{t}}}{T} \leq \frac{4R(1+C\kappa)\sqrt{\ln\left(\frac{12T}{\delta}\right)}}{\sqrt{T}},$$

where the last inequality holds since $\sum_{t=1}^{T}\frac{1}{\sqrt{t}} \leq 2\sqrt{T}$. Combining these inequalities together, we have with probability at least $1-\delta$, we have

$$Q(\bar{\theta}_T) - Q(\theta_*) \leq \frac{R^2}{\gamma T} + 10\gamma + \frac{R(20+4C\kappa)\sqrt{\ln(12T/\delta)}}{\sqrt{T}}.$$

Choosing $\gamma = \frac{R}{\sqrt{10T}}$, we have

$$Q(\bar{\theta}_T) - Q(\theta_*) \leq \frac{\left(2\sqrt{10} + (20+4C\kappa)\sqrt{\ln(12T/\delta)}\right)R}{\sqrt{T}}.$$

$\square$

**Lemma 3.** *With probability at least* $1-\delta$,

$$\sup_{\theta\in[0,0.5],\mathbf{z}\in\mathcal{Z}}\|\widehat{G}_t(\theta,\mathbf{z})-G(\theta,\mathbf{z})\|_2 \leq (1+C\kappa)\sqrt{\frac{\ln(4/\delta)}{t}}.$$

*Proof.* For any $\theta$ and any $\mathbf{z}$, the following argument holds. By Hoeffding's inequality, we have with probability at least $1-\frac{\delta}{2}$,

$$|\widehat{\pi}_t - \pi| \leq \sqrt{\frac{\ln(4/\delta)}{2t}}.$$

By the Assumption 1, we have with probability at least $1-\frac{\delta}{2}$,

$$|\hat{\eta}_t(\mathbf{x}) - \eta(\mathbf{x})| \leq C\kappa\sqrt{\frac{\ln(4/\delta)}{t}}.$$

Note that $0 \leq \theta \leq 0.5$, and hence we know that with probability at least $1-\delta$,

$$\mathrm{LHS} \leq |\widehat{\pi}_t - \pi|\cdot\theta + |\hat{\eta}_t(\mathbf{x}_t) - \eta(\mathbf{x}_t)| \leq (1+C\kappa)\sqrt{\frac{\ln(4/\delta)}{t}}.$$

$\square$

## 2.5 Proof of Theorem 3

Given Theorem 2, the proof of Theorem 3 follows similar as the analysis [1] by noting that the objective function $Q(\theta)$ is strongly convex which is a special case of uniformly convex. For completeness, we give a proof here.

*Proof.* Define

$$\bar{\delta} = \frac{2\delta}{\log_2 n}, \quad a(n, \bar{\delta}) = \frac{2\sqrt{10} + (20 + 4C\kappa)\sqrt{\ln(12n/\bar{\delta})}}{\sqrt{n}}.$$

$$\mu_0 = \frac{2a(n_0, \bar{\delta})}{R_0}, \quad \mu_k = 2^k\mu_0, \quad R_k = R_0/2^k$$

where $k = 1, \ldots, m$. Then we have $\mu_k R_k^2 = 2^{-k}\mu_0 R_0^2$.

By definition of $m$ in Algorithm 1 (FOFO), when $n \geq 100$,

$$0 < \frac{1}{2}\log_2\frac{2n}{\log_2 n} - 2 \leq m \leq \frac{1}{2}\log_2\frac{2n}{\log_2 n} - 1 \leq \frac{1}{2}\log_2 n, \tag{12}$$

so we have

$$2^m \geq \frac{1}{4}\sqrt{\frac{2n}{\log_2 n}}. \tag{13}$$

Define $c = \sqrt{\frac{2}{\sigma}}$, and note that $Q(\theta)$ is $\sigma$-strongly convex, and hence $\|\theta - \theta^*\|_2 \leq c(Q(\theta) - Q(\theta^*))^{\frac{1}{2}}$, where $\theta^*$ is the closest point to $\theta$ in $[0, 0.5]$.

Without loss of generality, we assume $c^2 \geq \frac{R_0}{2}$, i.e., $\frac{1}{c^2} \leq \frac{2}{R_0}$. Now we prove that $\frac{2}{R_0} \leq \mu_m$. When $n \geq 100$, we have

$$\mu_m = 2^m\mu_0$$

$$\geq \frac{1}{4}\sqrt{\frac{2n}{\log_2 n}}\frac{4}{R_0}\left(\frac{\sqrt{10}}{\sqrt{n_0}} + \frac{(10 + 2C\kappa)\sqrt{\ln(12n_0/\bar{\delta})}}{\sqrt{n_0}}\right)$$

$$\geq \frac{2}{R_0}\cdot\frac{1}{2}\sqrt{\frac{2n}{\log_2 n}}\left(\frac{\sqrt{10}}{\sqrt{n_0}} + \frac{8\sqrt{\ln(6\log_2 n)}}{\sqrt{n_0}}\right)$$

$$\geq \frac{2}{R_0}\sqrt{\frac{2n}{\log_2 n}}\sqrt{\frac{(8\sqrt{10})\sqrt{\ln(6\log_2 n)}}{n_0}}$$

$$\geq \frac{2}{R_0}\cdot\sqrt{\frac{2n}{\log_2 n}}\sqrt{\frac{(8\sqrt{10})\sqrt{\ln(3\log_2 n)}}{\frac{n}{m}}}$$

$$\geq \frac{2}{R_0}\cdot\sqrt{\frac{2n}{\log_2 n}}\sqrt{\frac{(8\sqrt{10})\sqrt{\ln(3\log_2 n)}}{\frac{n}{\frac{1}{2}\log_2\frac{2n}{\log_2 n} - 2}}}$$

$$= \frac{2}{R_0}\sqrt{(8\sqrt{10})\sqrt{\ln(3\log_2 n)}\left(1 - \frac{\log_2\log_2 n + 3}{\log_2 n}\right)}$$

$$\geq \frac{2}{R_0}.$$

where the first inequality holds because of (13), the second inequality stems from the fact that $10 + 2C\kappa > 8$, $0 < \delta < 1$, $n_0 \geq 1$, and the definition of $\bar{\delta}$, the third inequality holds by employing $a + b \geq 2\sqrt{ab}$, the fourth inequality holds because $0 < n_0 = \lfloor n/m \rfloor \leq n/m$, the fifth inequality holds because of the lower bound of $m$ in (12), and the last inequality holds since when $n \geq 100$, the function $(8\sqrt{10})\sqrt{\ln(3\log_2 n)}\left(1 - \frac{\log_2\log_2 n + 3}{\log_2 n}\right)$ is monotonically increasing with respect to $n$, and hence is greater than 1. So $\frac{2}{R_0} \leq \mu_m$. Recall that $\frac{1}{c^2} \leq \frac{2}{R_0}$, and thus, $\frac{1}{c^2} \leq \mu_m$.

Given $\widehat{\theta}_k$, denote $\widehat{\theta}_k^*$ by the closest optimal solution to $\widehat{\theta}_k$. We consider two cases.

**Case 1.** If $\frac{1}{c^2} \geq \mu_0$, then $\mu_0 \leq \frac{1}{c^2} \leq \mu_m$. So there exists a $k^*$ such that $\mu_{k^*} \leq \frac{1}{c^2} \leq \mu_{k^*+1} = 2\mu_{k^*}$, where $0 \leq k^* < m$. To utilize this fact, we have the following lemma.

**Lemma 4.** *Let $k^*$ satisfy $\mu_{k^*} \leq \frac{1}{c^2} \leq 2\mu_{k^*}$. Then for any $1 \leq k \leq k^*$, there exists a Borel set $\mathcal{A}_k \subset \Omega$ of probability at least $1 - k\bar{\delta}$, such that for $\omega \in \mathcal{A}_k$, the points $\{\widehat{\theta}_k\}_{k=1}^m$ generated by the Algorithm 1 satisfy*

$$\|\widehat{\theta}_{k-1} - \widehat{\theta}^*_{k-1}\|_2 \leq R_{k-1} = 2^{-k+1}R_0, \tag{14}$$

$$Q(\widehat{\theta}_k) - Q_* \leq \mu_k R_k^2 = 2^{-k}\mu_0 R_0^2. \tag{15}$$

*Moreover, for $k > k^*$ there is a Borel set $\mathcal{C}_k \subset \Omega$ of probability at least $1 - (k - k^*)\bar{\delta}$ such that on $\mathcal{C}_k$, we have*

$$Q(\widehat{\theta}_k) - Q(\widehat{\theta}_{k^*}) \leq \mu_{k^*} R_{k^*}^2. \tag{16}$$

*Proof.* We prove (14) and (15) by induction. Note that (14) holds for $k = 1$. Assume it is true for some $k > 1$ on $\mathcal{A}_{k-1}$. According to the Theorem 2, there exists a Borel set $\mathcal{B}_k$ with $\Pr(\mathcal{B}_k) \geq 1 - \bar{\delta}$ such that

$$Q(\widehat{\theta}_k) - Q_* \leq R_{k-1}a(n_0, \bar{\delta}) = \frac{1}{2}\mu_k 2^{-k}R_0 R_{k-1} = \mu_k R_k^2,$$

which is (15). By the inductive hypothesis, $\|\widehat{\theta}_{k-1} - \widehat{\theta}^*_{k-1}\|_2 \leq R_{k-1}$ on the set $\mathcal{A}_{k-1}$. Define $\mathcal{A}_k = \mathcal{A}_{k-1} \cap \mathcal{B}_k$. Note that

$$\Pr(\mathcal{A}_k) \geq \Pr(\mathcal{A}_{k-1}) + \Pr(\mathcal{B}_k) - 1 \geq 1 - k\bar{\delta},$$

and on $\mathcal{A}_k$, by the strong-convexity of $Q(\theta)$ and the definition of $k^*$, we have

$$\|\widehat{\theta}_k - \widehat{\theta}^*_k\|_2^2 \leq c^2(Q(\widehat{\theta}_k) - Q_*) \leq \frac{Q(\widehat{\theta}_k) - Q_*}{\mu_{k^*}} \leq \frac{\mu_k R_k^2}{\mu_{k^*}} \leq R_k^2,$$

which is (14) for $k + 1$.

Now we prove (16). For $k > k^*$, one can apply the similar strategy as in Theorem 2. Specifically, at the $k$-th stage with $k > k^*$, employing the similar proof of Theorem 2 by substituting all $\theta_*$ to $\widehat{\theta}_{k-1}$, the first term of RHS becomes zero and hence we get a tighter bound of $Q(\widehat{\theta}_k) - Q(\widehat{\theta}_{k-1})$, we here relax the bound to be $R_{k-1}a(n_0, \bar{\delta})$.

So there exists a Borel set $\mathcal{B}_k$ with $\Pr(\mathcal{B}_k) \geq 1 - \bar{\delta}$ such that

$$Q(\widehat{\theta}_k) - Q(\widehat{\theta}_{k-1}) \leq R_{k-1}a(n_0, \bar{\delta}) = 2^{k^*-k}R_{k^*-1}a(n_0, \bar{\delta}) = 2^{k^*-k}\mu_{k^*}R_{k^*}^2 = \mu_k R_k^2,$$

which implies that on $\mathcal{C}_k = \cap_{j=k^*+1}^k \mathcal{B}_j$, we have

$$Q(\widehat{\theta}_k) - Q(\widehat{\theta}_{k^*}) = \sum_{j=k^*+1}^k \left( Q(\widehat{\theta}_j) - Q(\widehat{\theta}_{j-1}) \right) \leq \sum_{j=k^*+1}^k 2^{k^*-j}\mu_{k^*}R_{k^*}^2 \leq \mu_{k^*}R_{k^*}^2.$$

By union bound, we have $\Pr(\mathcal{C}_k) = \Pr(\cap_{j=k^*+1}^k \mathcal{B}_j) \geq 1 - (k - k^*)\bar{\delta}$. Here completes the proof. $\square$

Now we proceed the proof as follows. Note that $\mu_0 \leq \frac{1}{c^2} \leq \mu_m$. At the end of $k^*$-th stage, on the Borel set $\mathcal{A}_{k^*}$ of probability at least $1 - k^*\bar{\delta}$, we have

$$Q(\widehat{\theta}_{k^*}) - Q_* \leq \mu_{k^*}R_{k^*}^2.$$

Then on the Borel set $\mathcal{D}_m = \mathcal{C}_m \cap \mathcal{A}_{k^*} = (\cap_{j=k^*+1}^m \mathcal{B}_j) \cap \mathcal{A}_{k^*}$ with $\Pr(\mathcal{D}_m) \geq 1 - m\bar{\delta}$, we have

$$Q(\widehat{\theta}_m) - Q_* = Q(\widehat{\theta}_m) - Q(\widehat{\theta}_{k^*}) + (Q(\widehat{\theta}_{k^*}) - Q_*) \leq 2\mu_{k^*}R_{k^*}^2 \leq 4(\frac{\mu_{k^*}}{c^{-2}})\mu_{k^*}R_{k^*}^2$$

$$= (4c \cdot a(n_0, \bar{\delta}))^2.$$

By the definition of $m$ and $\bar{\delta}$, and the fact that $m \leq \frac{1}{2}\log_2 n$, we have $m\bar{\delta} \leq \delta$. So $\Pr(\mathcal{D}_m) \geq 1 - \delta$.

Table 1: Offline Testing F-measure (bold numbers represent the best performance)

| Datasets | FOFO | OFO | LR | STAMP | OMCSL |
|---|---|---|---|---|---|
| webspam | $\mathbf{.9348 \pm .0003}$ | $.9348 \pm .0004$ | $.9347 \pm .0005$ | $.9312 \pm .0014$ | $.9282 \pm .0046$ |
| a9a | $\mathbf{.6789 \pm .0015}$ | $.6755 \pm .0020$ | $.6518 \pm .0026$ | $.6735 \pm .0034$ | $.6704 \pm .0096$ |
| ijcnn1 | $\mathbf{.6412 \pm .0020}$ | $.5776 \pm .0039$ | $.4441 \pm .0040$ | $.5987 \pm .0328$ | $.6050 \pm .0225$ |
| w8a | $\mathbf{.7159 \pm .0118}$ | $.6695 \pm .0134$ | $.6621 \pm .0222$ | $.6706 \pm .0289$ | $.6627 \pm .0370$ |
| covtype (2 vs o) | $\mathbf{.7627 \pm .0005}$ | $.7625 \pm .0005$ | $.7557 \pm .0004$ | $.7568 \pm .0055$ | $.7557 \pm .0081$ |
| covtype (1 vs o) | $\mathbf{.7090 \pm .0004}$ | $.7082 \pm .0002$ | $.6770 \pm .0010$ | $.7039 \pm .0047$ | $.7000 \pm .0093$ |
| cov (3 vs o) | $\mathbf{.7277 \pm .0009}$ | $.7257 \pm .0005$ | $.6914 \pm .0039$ | $.7213 \pm .0050$ | $.7210 \pm .0050$ |
| covtype (7 vs o) | $\mathbf{.6723 \pm .0022}$ | $.6521 \pm .0025$ | $.6140 \pm .0037$ | $.6417 \pm .0197$ | $.6513 \pm .0150$ |
| covtype (6 vs o) | $\mathbf{.4468 \pm .0015}$ | $.4251 \pm .0014$ | $.1258 \pm .0072$ | $.3971 \pm .0516$ | $.4237 \pm .0142$ |
| covtype (5 vs o) | $\mathbf{.2648 \pm .0036}$ | $.2488 \pm .0027$ | $.0000 \pm .0000$ | $.2218 \pm .0246$ | $.2362 \pm .0304$ |
| covtype (4 vs o) | $\mathbf{.5512 \pm .0035}$ | $.5228 \pm .0083$ | $.4123 \pm .0130$ | $.3682 \pm .0724$ | $.5139 \pm .0256$ |
| Sensorless (1 vs o) | $\mathbf{.7549 \pm .0047}$ | $.6732 \pm .0022$ | $.4774 \pm .0156$ | $.6243 \pm .1394$ | $.5401 \pm .2360$ |
| Sensorless (2 vs o) | $\mathbf{.4698 \pm .0178}$ | $.2388 \pm .0083$ | $.1667 \pm .0000$ | $.3284 \pm .1485$ | $.4689 \pm .0330$ |
| Sensorless (3 vs o) | $.2138 \pm .0047$ | $\mathbf{.2254 \pm .0048}$ | $.1345 \pm .0709$ | $.1819 \pm .0812$ | $.1804 \pm .0413$ |
| Sensorless (4 vs o) | $\mathbf{.5895 \pm .0055}$ | $.3117 \pm .0102$ | $.1360 \pm .0717$ | $.3778 \pm .2152$ | $.4530 \pm .0813$ |
| Sensorless (5 vs o) | $\mathbf{.3089 \pm .0049}$ | $.2343 \pm .0047$ | $.1009 \pm .0868$ | $.2264 \pm .1186$ | $.1782 \pm .1228$ |
| Sensorless (6 vs o) | $\mathbf{.3607 \pm .0062}$ | $.2789 \pm .0078$ | $.0993 \pm .0854$ | $.2772 \pm .0702$ | $.2266 \pm .1503$ |
| Sensorless (7 vs o) | $.9994 \pm .0002$ | $\mathbf{.9996 \pm .0001}$ | $.9986 \pm .0010$ | $.9988 \pm .0009$ | $.9982 \pm .0017$ |
| Sensorless (8 vs o) | $\mathbf{.4085 \pm .0017}$ | $.3158 \pm .0047$ | $.0496 \pm .0799$ | $.3185 \pm .1159$ | $.3484 \pm .0583$ |
| Sensorless (9 vs o) | $\mathbf{.2783 \pm .0037}$ | $.2069 \pm .0039$ | $.1346 \pm .0710$ | $.1749 \pm .1352$ | $.1902 \pm .1251$ |
| Sensorless (10 vs o) | $\mathbf{.6025 \pm .0080}$ | $.4897 \pm .0113$ | $.1659 \pm .0000$ | $.4089 \pm .2345$ | $.5170 \pm .0566$ |
| Sensorless (11 vs o) | $.9997 \pm .0000$ | $.9997 \pm .0002$ | $.9998 \pm .0002$ | $.9997 \pm .0001$ | $\mathbf{.9998 \pm .0002}$ |
| protein (1 vs o) | $.5008 \pm .0026$ | $\mathbf{.5037 \pm .0059}$ | $.4643 \pm .0114$ | $.4914 \pm .0163$ | $.4930 \pm .0116$ |
| protein (2 vs o) | $\mathbf{.6849 \pm .0035}$ | $.6835 \pm .0040$ | $.6390 \pm .0053$ | $.6787 \pm .0069$ | $.6735 \pm .0144$ |
| protein (0 vs o) | $.7479 \pm .0017$ | $\mathbf{.7483 \pm .0014}$ | $.7183 \pm .0023$ | $.7430 \pm .0071$ | $.7423 \pm .0052$ |

**Case 2.** If $\frac{1}{c^2} < \mu_0$, then on $\mathcal{A}_1 = \mathcal{B}_1$,

$$Q(\widehat{\theta}_1) - Q_* \leq R_0 \cdot a(n_0, \bar{\delta}) = \frac{R_0}{a(n_0, \bar{\delta})} \cdot a(n_0, \bar{\delta})^2 = \frac{2}{\mu_0} a(n_0, \bar{\delta})^2 \leq 2 \left( c \cdot a(n_0, \bar{\delta}) \right)^2.$$

Hence on $\mathcal{A}_1 \cap \mathcal{C}_m$, by using Lemma 4 and a similar argument as in case 1, we have

$$Q(\widehat{\theta}_m) - Q_* = Q(\widehat{\theta}_m) - Q(\widehat{\theta}_1) + Q(\widehat{\theta}_1) - Q_* \leq 2R_0 \cdot a(n_0, \bar{\delta}) \leq (2c \cdot a(n_0, \bar{\delta}))^2,$$

where $\Pr(\mathcal{A}_1 \cap \mathcal{C}_m) \geq 1 - \delta$.

Combining the two cases, we have with probability at least $1 - \delta$,

$$Q(\widehat{\theta}_m) - Q_* \leq (4c \vee 2c)^2 \left( a(n_0, \bar{\delta}) \right)^2 = \widetilde{O}\left( \frac{\ln(\frac{1}{\delta})}{\sigma n} \right).$$

$\square$

## 3 More Experimental Results

More experimental results are reported in Table 1 (offline testing results) and Figure 1 (online F-measure vs running time).