[Reviews · NeurIPS 2018]

Reviewer 1



A) My main concern with this paper is with respect to the main results (Theorems 2 and 3). It seems the authors have not put sufficient care to the fact that \partial \hat{Q} in Algorithm 2 is a biased estimator of the true gradient \partial Q. Also, \hat{Q} defined in Line 189 depends on \hat{\pi} which is an estimate of \pi. Thus, a probabilistic proof would require to look at a conditional probability of the estimation of Q depending on the estimation of \pi. Assumption 1 somehow disregards this. B) Regardless of the above, the final high probability statement in Theorems 2 and 3, seem to be missing the union bound of the error probability in Assumption 1. That is, Algorithm 2 calls \partial \hat{Q} a total of T times. C) I let the authors know that there is some literature on convergence rates using biased gradient estimators. For instance, in Section 3 of [Honorio, "Convergence Rates of Biased Stochastic Optimization for Learning Sparse Ising Models". ICML 2012] results are given for general Lipschitz functions and samplers with decaying bias and variance. Similarly, [Hu et al, "(Bandit) Convex Optimization with Biased Noisy Gradient Oracles". AISTATS 2016] analyzes a general regime. I wonder whether the current problem fits or not the above mentioned frameworks. === AFTER REBUTTAL === Regarding A, I thank the authors for the clarification. After inspecting Lemma 3, everything seems clear. Regarding B, I am satisfied with the authors' response. Although the authors should notice that Algorithm 2 calls the random part T times, and Algorithm 1 calls Algorithm 2 m times, thus I believe the final result with respect to Algorithm 1 requires an additional log m term. Regarding C, while I partially disagree with the authors, I am not basing my decision on this point. After inspection, I believe Assumption 1 is somehow misleading, as it hides dependence with dimension d. Let me elaborate a bit. Intuitively speaking w^* is the vector that we would learn by using logistic regression from infinite data (Thus, approaching the expectation with respect to the data distribution). Then w_t is what we would learn from t data points from the unknown data distribution. I believe the only assumption is boundedness (Line 208 in manuscript, Line 72 in proof of appendix). After that, Lipschitz continuity of logistic regression allows to bound \eta pointwise. Assumption 1 (w_t approaching w^* in L2 norm) is a bit misleading. As Reviewer 4 points out, in general one could get O(1/\sqrt{t}) generalization bounds for the risks, but getting convergence in L2 norm requires more restrictions. One piece of literature that I can recall is the restricted strong convexity framework. In this case, some restrictive data distribution assumptions are necessary, such as: data being produced by a logistic regression model. Also importantly, without any assumption on simplicity of representation of w^* (e.g., sparsity) the constant C' in Assumption 1 will very likely depend on the dimension d, since the authors are bounding the L2 norm. This happens for several learning theoretic methods: VC dimension, Rademacher complexity, primal-dual witness and restricted strong convexity. The above makes Lemma 3 and Theorem 2 dependent on data dimension, thus, obtaining O(d/\sqrt{T}).

Reviewer 2



Summary: Direct optimizing F-measure has gotten constant attention over time, and authors extends the latest online F-measure optimization (OFO), and proposes a new method called FOFO (Fast OFO). Albeit the fact the proposed algorithm has a similar asymptotic computation time and memory consumption with OFO, authors proves the convergence rate bound, which didn’t exist for the previous study. Authors also conducted various experiments to prove their algorithm. Quality: The technical content of the paper appears to be solid with various theoretical analyses and with experiments. I liked the in-depth related work section comparing various previous F1-measure optimization-approaches against this work. One thing is that, I would like to see more direct comparison of the OFO/FOFO in the algorithm section. In particular, additional convex optimization problem was introduced in FOFO compared to OFO, so authors might remark this fact when they were describing the algorithm. Also, I would like to encourage authors to include time-vs-F1-score figures in the main paper (as their main contribution is fast learning) and also add an additional memory consumption comparison experiments. Clarity: The paper is pretty readable, and well structured. I did not find trouble following the main contributions and supporting materials of the paper. Originality: The main contribution of this paper is to suggest a new algorithm with additional theoretical convergence guarantee. This made possible by changing the way they choose the optimal threshold by solving a convex problem instead of choosing a heuristic way (previous work). Even though the paper seems to be a direct extension of a previous method (OFO), the paper has original materials such as new theoretical analysis. Significance: The paper seems to be a useful contribution to various applications to maximize the F1 measures. In particular, their experimental results show consistent improvements over other methods. Conclusion: I believe this is a solid paper, probably not the best paper though. -------------------------------------------------------------------------------------------- Post rebuttal: I thank authors for the clarifications. I have read other reviewer's responses and discussions and decided to keep the current score believing authors handle the remaining concerns made by reviewer #2 and #4 in their final version of the paper. Thank you.

Reviewer 3



The paper proposes a stochastic/online algorithm for optimizing the F-measure. Previous stochastic approaches for F-measure optimization either come with guarantees of rates of convergence to the optimal value of a 'surrogate' objective, or come with asymptotic consistency guarantees for the 'original' F-measure. In this paper, the authors build on ideas from Busa-Fekete et al. (2015) and develop an algorithm which has better empirical performance than previous methods, while enjoying (non-asymptotic) rates of convergence to the optimal F-measure value. The proposed algorithm is based on the observation that the optimal classifier for the F-measure can be obtained by thresholding the posterior class probability at a suitable point, and performs simultaneous stochastic updates on the posterior class probability model and on the threshold. While Busa-Fekete et al. pose the threshold selection problem as a root finding problem, the present paper formulates this problem as a strongly convex optimization problem. This allows the authors to show that under very specific assumptions on the posterior class probability model and on the algorithm for learning the model, the proposed approach has a O(1/\sqrt{n}) convergence rate to the optimal F-measure value. Pros: - The idea of posing the threshold selection problem as a strongly convex optimization problem is indeed interesting. - The proposed algorithm seems to empirically out-perform the state-of-the-art algorithms for F-measure, achieving higher or comparable F-measure values at a faster rate. Cons: However, the convergence/consistency guarantee that the authors provide seems to be under strong assumptions on the posterior class probability distribution and on the algorithm used to learn the posterior probability model. In particular, the authors assume/state: - The posterior class probability distribution can be parametrized as a generalized linear model - The expected logistic loss computed on the posterior probability model is (globally) *strongly convex* - There exists an online algorithm that enjoys *high probability convergence rates* to the optimal parameters w^* of the posterior prob. model, i.e. can provides iterates w_1,\ldots, w_T such that for each ’t’, with high probability, ||w_t - w^*|| \leq C’ / t, for a problem-independent constant C’ (of these the second statement is not an assumption, but stated in ll. 217) I find the above to be problematic: Firstly, the assumption that the posterior class probability distribution takes a very specific parametrized form seems restrictive, especially because the analysis gives no room for model misspecification. I understand the analysis might also extend to slightly more general posterior probability distributions that can be written as a sigmoid of a score function, but even this is a special case, and excludes a whole range of real-world distributions. Secondly, the authors’ statement that the expected logistic loss is (globally) strongly convex is not true in general. The authors cite Agarwal et al. (2012) [1] , but even there it is only shown that the expected logistic loss is locally strongly convex (p.6) where the strong convexity parameter depends on the marginal distribution over ‘x’ (in particular on the min. Eigen value of the covariance matrix of ‘x’). Finally, in the absence of (global) strong convexity, I do not think (as the authors claim) the algorithm of Agarwal et al. (2012) [15] can be applied to provide high probability convergence guarantees on individual model vectors. Indeed there have been some works (e.g. Bach et al., "Adaptivity of Averaged Stochastic Gradient Descent to Local Strong Convexity for Logistic Regression”, JMLR’14) that analyze SGD specifically for the logistic loss, but seem to provide weaker “expected” guarantees (and where C’ would be distribution-dependent) (also please see pp. 2, end of paragraph 2 in Bach (2014) for a discussion on strong convexity of logistic loss) It would be good if the authors are able to provide a reference to a specific algorithm that can satisfy the assumptions they make. I do understand that the previous work of Busa-Fekete et al. also makes assumptions on the class probability learner, but their assumptions was on the class probability estimates \eta(x) and in particular on the expected absolute difference between the current class probability estimates and the true class probabilities (i.e. on E[|\eta(x) - \hat{\eta}_t(x)|]). The present paper makes stronger assumptions, namely a specific parametric form for the underlying class probability model, and high probability rates of convergence on the model parameters. Having said this, I would also like to mention that I am sympathetic to this line of work and can see why it is challenging to provide non-asymptotic guarantees for F-measure optimization (i.e. on the original F-measure and not a surrogate). The strongly convex re-formulation that the authors propose for threshold search is definitely interesting, and their empirical results indicate that this formulation can be beneficial in practice. However, the theoretical guarantees provided seem to be under restrictive assumptions. Other Comments: - Experiments: -- Please report F-measure performance as a function of running time rather than *iteration count*. Since an iteration may constitute different computations for different methods, it is only fair that you compare the methods against their running time. Indeed you do include run-time plots in the supplementary material, but please do have them in the main text. -- I see a large gap between FOFO and OFO for some of the data sets. Do you find that when you run for enough iterations eventually, both FOFO and OFO converge to the same threshold? How does the learned threshold compare with an offline/brute-force threshold tuning procedure? -- I see that the STAMP method you compare against has an epoch size parameter. How do you tune this parameter? Is there a specific reason you chose the number of classifiers in OMCSL to 10? - Remark on line 232: You mention that Theorem 2 (convergence of inner SFO) can be used to derive a O(1/\sqrt{n}) convergence result for FIFO. I don’t see immediately how you would get this result (are you referring to having only one run of SFO with T = n?). In particular, Theorem 2 holds under the assumption that |\theta_1 - \theta^*| \leq R. This is true for the first invocation of SFO, but for subsequent invocations (k > 2), where the radius R is multiplicatively reduced, how do you make sure that this condition continues to hold? - Same sample used for learning probability model and threshold: It seems to me that in Algorithm 2, the same sample is used by for both updating the posterior model parameters w_t (line 15) and updating the threshold \theta_t (line 13). I wonder if this would introduce an additional probabilistic dependence in your analysis. My guess is that a dependence is avoided by first updating the threshold and then the posterior model. Minor comments: - Definition 1: Might be good to explicitly mention that the probability is over random draw of n samples - Lemma 2: It seem from the proof in the supplementary material that c = 2/\pi for all \theta. May want to just mention this in the lemma. - Theorem 3: SFO -> FOFO, also should ‘m’ be a superscript on \theta (looking at Algorithm 1) ------- POST REBUTTAL ------- ------------------------------------ I thank the authors for the rebuttal. Having read their response and heard the views of the other reviewers, I'm increasing my score to 7. I would however recommend that the authors: (1) restate assumption 1 in terms of a condition on the estimated class probabilities \hat{\eta}_t (rather than making a specific parameterized assumption on the posterior distribution) (2) give examples where this assumption holds (one of which is when the expected loss is strongly convex) (3) clarify up-front in the paper that their result holds under a ‘point-wise’ convergence assumption and that it differs from previous works (currently the authors just mention that they need a ‘mild’ condition) If one were to just assume that the estimated probabilities converge in a weaker expected sense (e.g. the assumption in [3[), would your algorithm at least have an asymptotic convergence guarantee like [3]? If so, I would appreciate it if you could add a discussion on this.

Reviewer 4



Summary of the paper -------------------- The paper considers online F-measure optimization (OFO). While the previous approaches (except for [3]) either make use of convex surrogates for F-measure to achieve convergence or turn the problem into binary cost-sensitive classification, the reviewed work optimizes F-measure directly by simultaneously estimating the conditional class distribution \eta(x) = P(y=1|x), and the optimal threshold \theta on \eta to eventually converge to the population maximizer of F-measure (which is guaranteed by the past work to be a threshold function on \eta). The most related reference is [3], in which a similar procedure is undertaken. The main difference between [3] and the reviewed work is that in [3] the current threshold is simply set to F/2, a property satisfied by the optimal classifier based on true \eta, but not necessarily by the classifier based on the estimate \hat{\eta}; whereas the current paper uses a different method for estimating the threshold, which is based on stochastic gradient descent. The main advantage of the current work is that [3] only gives asymptotic convergence guarantee, while here the authors are able to show O(1/\sqrt{n}) converge rate with high probability. This advantage is further confirmed in the experimental study, where it is shown that the proposed algorithm is competitive or outperforms all existing approaches, including the one from [3]. The analysis in the paper makes use of the recent advances in the analysis of stochastic algorithms for strongly convex objectives where the strong convexity constant is unknown [8]. The idea is, given n samples, to split them into m = O(log(n)) phases, and in each phase to run a standard SGD algorithm on T=n/m samples with O(1/sqrt(T)) learning rate, but shrink the set of parameters to a ball of exponentially (w.r.t. to the number of phases) decreasing radius at the previous parameter. This approach guarantees fast O(1/n) converge to the optimum without knowing the strong convexity constant. Overall evaluation ------------------ I found the paper to be interesting and contributing significantly to the current state of online optimization of composite objective functions for binary classification. The main novelty is dealing with simultaneous estimation of class conditional distribution and optimization of the threshold, which requires handling gradient estimates which are biased and optimizing a function with unknown strong convexity constant. The main drawback is that the paper makes very strong assumption about the underlying probabilistic model, i.e. that the true conditional distribution \eta(x) = P(y=1|x) comes from a generalized linear model and can be computed as a logistic function of the dot product w^T x (the model is "well-specified"). From what I understand, this assumption can be relaxed to having some underlying model \eta(x) Lipschitz in w under the condition that an online algorithm is available which is guaranteed to converge to w^* at the rate O(1/sqrt(T)), where w^* is the weight vector which give rise to the true conditional distribution eta(x). This is, however, still a constraining assumptions, unlikely to be met in practice. I suggest to overally improve the language, as there are many typos and language mistakes (some of which are mentioned below). Also, the presentation of the proof should be made more readable. I checked all the proofs in the appendix and they seem to be correct, with several exceptions mentioned below. Despite the above shortcomings, I lean towards accepting the paper. Technical remarks ----------------- Lemma 2: The constant in the lemma is distribution-dependent, which should be clearly stated (it depends on \pi, the positive class prior). Assumption 1 and discussion thereafter: - Assumption 1 might be impossible to satisfy in practice, even under well-specified model assumption made in this paper. This is because for some distributions on a compact support, convergence of L(w) to L(w^*) does not imply that ||w-w^*|| converges to zero (e.g., when the distribution is concentrated on a single x, then all w such that w^T x = w*^T x give the optimal value of L(w)). In fact, w^* may not be uniquely defined. - (related to the previous remarks) the objective L(w) is not necessarily strongly convex, contrary to what is being claimed. Indeed, if the distribution of x is degenerate and concentrated on a single x value, the Hessian of L(w) is a rank one matrix. Proof of Theorem 2: Bounding the term IV -- Lemma 3 is used separately in each trial t (with probability at least 1-\delta_t), so by the union bound, the upper bound on IV holds with probability at least 1 - \sum_t \delta_t, not with probability 1 - \delta/3 as claimed (unless a stronger version of Hoeffding would be used with also holds over all t). Despite devoting some time into the proof of Theorem 3, I was unable to check its correctness in detail, as the proofs seems to be directly adapted from [8] and does not not give much guidance to the reader. Minor remarks ------------- - line 26: "this is because that" -> "this is because" - 39: depends -> depend - 93: "In order for online prediction" -- ? - 95: "predication" -> "prediction" - 126: Sine -> Since - Algorithm 1, line 4: ,, -> , - Algorithm 2: what is \theta_1 in the first line? - line 213: "in high probability" -> "with high probability" - Theorem 3: "Convergence Result of SFO" -> "Convergence Result of FOFO" (?) - Supplement, Eq. between lines 52-53: T is missing in the denominator in the first term on the right-hand side; also, \theta should be \theta^* on the left-hand side? - Supplement, Eq. (9) is missing -------------------- I have read the rebuttal, and two issues related to strong convexity and union bound were completely clarified by the authors, thank you.